# Psychometric properties of performance-based measures of physical function administered via telehealth among people with chronic conditions: A systematic review

Caoimhe Barry Walsh[1]*, Roisin Cahalan[1,2], Rana S. Hinman[3], Kieran O' Sullivan[1,4,5]

**1** School of Allied Health, University of Limerick, Limerick, Ireland, **2** Physical Activity for Health Centre, University of Limerick, Limerick, Ireland, **3** Centre for Health, Exercise and Sports Medicine, Department of Physiotherapy, University of Melbourne, Melbourne, Victoria, Australia, **4** Sports and Human Performance Centre, University of Limerick, Limerick, Ireland, **5** Ageing Research Centre, University of Limerick, Limerick, Ireland

* Caoimhe.Barry.Walsh@ul.ie

**Data Availability Statement:** All relevant data are within the manuscript and its Supporting Information files.

## Abstract

### Background

Telehealth could enhance rehabilitation for people with chronic health conditions. This review examined the psychometric properties of performance-based measures of physical function administered via telehealth among people with chronic health conditions using the Consensus-Based Standards for the Selection of Health Measurement Instruments (COSMIN) approach.

### Methods

This systematic review was registered with Prospero (Registration number: CRD42021262547). Four electronic databases were searched up to June 2022. Study quality was evaluated by two independent reviewers using the COSMIN risk of bias checklist. Measurement properties were rated by two independent reviewers in accordance with COSMIN guidance. Results were summarised according to the COSMIN approach and the modified GRADE approach was used to grade quality of the summarised evidence.

### Results

Five articles met the eligibility criteria. These included patients with Parkinson's Disease (n = 2), stroke (n = 1), cystic fibrosis (n = 1) and chronic heart failure (n = 1). Fifteen performance-based measures of physical function administered via videoconferencing were investigated, spanning measures of functional balance (n = 7), other measures of general functional capacity (n = 4), exercise capacity (n = 2), and functional strength (n = 2). Studies were conducted in Australia (n = 4) and the United States (n = 1). Reliability was reported for twelve measures, with all twelve demonstrating sufficient inter-rater and intra-rater reliability. Criterion validity for all fifteen measures was reported, with eight demonstrating sufficient validity and the remaining seven demonstrating indeterminate validity. No studies reported data on measurement error or responsiveness.

**Funding:** One author (CBW) is supported by a research studentship from The Faculty of Education and Health Sciences, University of Limerick, Ireland. RSH is supported by a National Health & Medical Research Council Senior Research Fellowship (#1154217).

**Competing interests:** The authors have declared that no competing interests exist.

## Conclusions

Several performance-based measures of physical function across the domains of exercise capacity, strength, balance and general functional capacity may have sufficient reliability and criterion validity when administered via telehealth. However, the evidence is of low-very low quality, reflecting the small number of studies conducted and the small sample sizes included in the studies. Future research is needed to explore the measurement error, responsiveness, interpretability and feasibility of these measures administered via telehealth.

## Introduction

Chronic health conditions have the potential to lead to significant levels of disability, mortality and reduced quality of life [1]. In 2019, on average, more than one-third of adults aged 16 and above in 26 OECD (Organisation for Economic Co-operation and Development) countries reported living with a chronic health condition [2]. The ageing nature of the Western world and the increasing prevalence of chronic conditions presents a significant socioeconomic burden and will continue to persistently challenge health care services [3, 4]. Rehabilitation has been identified as an integral aspect of chronic condition management, in order to facilitate people living with chronic health problems to independently manage their condition and improve their physical function and quality of life [5–8].

Although in-person rehabilitation is considered the default service delivery method, healthcare services lack the capacity required to meet the increasing demand for these programmes. Also, uptake levels among patients have traditionally been poor due to different barriers, such as travel and time limitations [9, 10]. Offering rehabilitation via digital platforms (telerehabilitation) may increase service accessibility and overcome barriers to traditional face-to-face programmes. Furthermore, telerehabilitation is as clinically effective as face-to-face rehabilitation for several different chronic populations [11–13]. The recent COVID-19 pandemic presented challenges for rehabilitation service providers, resulting in a dramatic increase in the use of telehealth. This accelerated shift towards the use of an alternative method of service delivery allowed health care services to maintain service accessibility and ensure continuity of patient care. Despite the evidence supporting its efficacy, resistance to the adoption of telehealth has been demonstrated by both patients and healthcare providers [14].

One of the challenges which has limited the adoption of telehealth is the perceived difficulty of assessing patients remotely, particularly the administration of performance-based measures via telehealth platforms and the uncertainty regarding the accuracy and reliability of these measures [15–17]. The use of standardised performance-based measures in clinical assessment is an important element of evidence-based rehabilitation and clinical practice [18, 19] to inform diagnosis, clinical decision making, intervention planning and goal setting [15, 20]. The regular measurement of parameters of performance-based physical function during rehabilitation programmes therefore facilitates objective monitoring and evaluation of the effectiveness of the intervention.

The reliability and validity of measures administered via telehealth has been explored in recent systematic, scoping and rapid reviews in musculoskeletal [15, 21], as well as chronic cardiac and respiratory [22, 23] populations. Zischke et al. [24] also conducted a review examining various clinical assessments conducted via telehealth. Overall, these reviews supported the feasibility of assessment via telehealth, and highlighted the reliability and validity of several

performance-based measures across domains such as range of motion, strength, endurance, aerobic capacity, balance, gait and functional assessments.

However, the existing evidence exploring performance-based measures is limited, with a tendency to focus on the use of measures in specific patient cohorts, rather than considering all domains across all populations with chronic conditions. Furthermore, some of the existing reviews included patient-reported outcomes such as pain intensity, or pain response during special orthopaedic tests. While evidence demonstrates that electronic patient-reported measures are equivalent to paper-based self-reported measures when administered in various chronic populations [25–28], there is limited evidence exploring the psychometric properties and equivalence of performance-based measures administered via telehealth when compared to face-to-face administration.

Therefore, a comprehensive overview of a wide range of performance-based measures relevant to a variety of chronic neurological, respiratory and musculoskeletal conditions administered via telehealth is required. To our knowledge, this is the first review using the Consensus-Based Standards for the Selection of Health Measurement Instruments (COSMIN) approach to evaluate the reliability and validity of performance-based measures of physical function across a broad range of chronic health conditions.

## Methods

This systematic review protocol was registered with Prospero (Registration number: CRD42021262547). This review was conducted in accordance with COSMIN methodology which is a robust approach that aims to improve the selection of measurement instruments using transparent methodology.

### Search strategy

A comprehensive search strategy was developed, reviewed and refined by the authors, with the assistance of a health librarian, in accordance with the Preferred Reporting Items for Systematic Reviews and Meta-Analyses (PRISMA) guidelines [29] (S1 Fig). An electronic database search of PubMed, EMBASE, CINAHL and PsycINFO via EBSCOhost was conducted on the 28th of June 2022. Key search terms were developed using four individual search filters. These filters included:

1. Population: chronic conditions OR chronic disease OR chronic health OR chronic illness OR long term illness OR long term disability OR long term condition

2. Construct: physical function OR physical performance OR functional capacity OR physical capacity

3. Measurement Instrument: assessment OR evaluation OR outcome OR measure OR test

4. Context: telehealth OR telerehabilitation OR telemedicine OR e-health

These individual filters were combined with the COSMIN search filter for measurement properties [30] to create the search strategy outlined in S2 Fig. Hand-searching of the reference lists of the included articles was also performed to identify additional relevant articles.

### Eligibility criteria

Studies were included in the review if they met the following criteria:

1. Population: adults (≥18 years) diagnosed with any chronic health condition, as defined by the ICD-10-CM [31] as a condition that lasts greater than 12 months and results in the

need for ongoing medical intervention and limits self-care, independent living and social interaction. Studies including a mixed sample of acute and chronic populations were included if at least 80% of the sample had a chronic diagnosis.

2. Construct: the evaluated measure was a performance-based measure of physical function, as defined by the World Health Organisation (WHO) [32] International Classification of Functioning, Disability and Health (ICF) framework as activities which relate to the ability to move around and perform daily activities e.g. strength, balance, etc.

3. Measurement instrument: an established performance-based measure of physical function, commonly used in clinical practice, which was evaluated synchronously by a tester as the activity was being performed by the individual. This usually involved evaluation by timing, counting or distance methods [33].

4. Setting: the evaluated measure was administered by a tester located remotely from the patient using any telehealth platform, as defined by WHO as "the delivery of health care services, where patients and providers are separated by distance. Telehealth uses information and communication technologies (ICT) for the exchange of information for the diagnosis and treatment of diseases and injuries.".

5. Measurement properties: In our pre-registered protocol, we highlighted studies must have reported one or more of the psychometric measurement properties from the COSMIN taxonomy [30]. For studies examining the validity of the measurement instrument administered via telehealth, the comparator was a face-to-face administration of the same measurement instrument. Since the comparator was always face-to-face administration of the same measure, when extracting data from the selected studies the measurement properties of interest were reliability, measurement error and criterion validity. Therefore, the remaining measurement properties outline in the COSMIN taxonomy including other forms of validity and interpretability were not considered to be outcomes of interest in this review.

Studies were excluded if (1) the evaluated measure was a self-reported measure of physical function, or a laboratory value (e.g., $VO_2$ max, spirometry, etc.) indirectly used to assess physical function, or a self-administered measure that did not involve administration and evaluation by an independent tester; or (2) the study population consisted of post-operative patients since post-operative pain and disability levels differ in magnitude and stability from chronic conditions.

A sample of 30% of abstracts from the database search were initially screened by two independent reviewers (CBW & RC) to determine potential eligibility. As good agreement (>80%) was achieved, the remaining abstracts were screened by one reviewer (CBW). Thereafter, a sample of 30% of full texts of potentially eligible studies were reviewed to determine eligibility by two independent reviewers (CBW & RC). Any disagreements were resolved through discussion with a third reviewer (KOS). As above, good agreement was achieved, and the remaining studies were reviewed by one reviewer (CBW).

## Data extraction

Data were extracted by two independent authors (CBW & KOS) using a table created by the authors following COSMIN guidance [34]. Firstly, the characteristics of the included studies and the performance-based measures evaluated within the studies were extracted. Thereafter, data relating to the evaluation of the methodological quality of the included studies and the evaluation of the measurement properties (i.e. strength of correlations/associations) were also

extracted. The included performance-based measures were categorised according to the domain of physical function that they measured. These domains included exercise capacity, functional strength, functional balance and general functional capacity.

## Methodological quality of included studies

The methodological quality of each of the included studies was evaluated by two independent authors (CBW & KOS) using the COSMIN risk of bias checklist and scores were determined by consensus [34]. This tool contains separate standards for each measurement property (i.e., reliability, measurement error and criterion validity) that can be used to determine the trustworthiness of the result. Each of the standards were rated and the 'worst-score-counts' method was applied to determine the overall quality of each measurement property reported in the included studies [34].

## Evaluation of the measurement properties reported in the included studies

The COSMIN methodology was used to evaluate the measurement property results reported in each of the included studies [34]. These results were evaluated according to the criteria for good measurement properties (strength of correlations/associations with the reference standard face-to-face administration of the measure) to give a rating of sufficient (+), indeterminate (?) or insufficient (-) for each measurement property, as described by Prinsen *et al.* [34] (See S1 Table).

For the reliability domain, inter-rater and intra-rater reliability were evaluated by comparing the scores for the measure when administered via telehealth between different raters and also when administered by the same rater at two different time points. As recommended, a threshold of 0.70 on the intraclass correlation (ICC) or weighted kappa was used to evaluate the reliability of the measure administered via telehealth [34]. If the correlation was $\geq 0.70$ the reliability received a sufficient rating. If the ICC or weighted kappa was not reported it received an indeterminate rating for reliability. The reliability of the measure was rated as insufficient if the ICC or weighted kappa score was $< 0.70$.

For the criterion validity domain, the measure administered via telehealth was compared to the same measure administered in a face-to-face environment. A correlation of 0.70 with the reference standard [34], which for the purpose of this review was the measure administered in a face-to-face environment, was the threshold. The validity of the measure was rated as sufficient if the correlation was $\geq 0.70$. The validity was rated as indeterminate if correlations were not reported. The validity was rated as insufficient if the correlation with the reference standard was $< 0.70$.

## Data synthesis and analysis

To synthesise the results, the evidence was summarised per measurement property (e.g., reliability, validity) per outcome measure to come to an overall conclusion regarding the reliability and validity of the measures. If multiple studies examined the same measure, the results of the studies were synthesised to achieve an overall result. In the case of inconsistency in the results between studies (e.g., both sufficient and insufficient results were found), explanations for the inconsistency were explored. When inconsistent results likely existed due to varying study quality as previously described, the results of lower quality studies were omitted and only the higher quality results were used to determine the overall rating and the quality of summarised evidence was downgraded due to inconsistency. If no logical explanation was found which could explain the inconsistency, the results were considered inconsistent. The modified GRADE approach was used by two independent reviewers and disagreements were resolved

by consensus to summarise how confident we can be that the summarised evidence is trustworthy [35]. The summarised evidence was graded as high, moderate, low or very low based on the following four criteria: 1. Risk of Bias (quality of the studies); 2. Inconsistency (of the results of the studies); 3. Imprecision (total sample size of the studies) and 4. Indirectness (evidence from different populations than the population of interest). Detailed instructions on the use of the modified GRADE approach to grade the quality of the summarised evidence can be found in S2 Table. The starting point assumed that the summarised result was of high quality and was downgraded by one, two or three levels depending on the risk of bias. The summarised result was further downgraded depending on the inconsistency, imprecision and indirectness associated with the summarised result as appropriate.

When inconsistency existed between the results of the included studies examining the same measurement instrument, the results were summarised as sufficient or insufficient and the quality of the evidence was downgraded for inconsistency with one or two levels depending on the severity of the inconsistency. As the severity of inconsistency between results is context dependent, the level of severity was discussed and decided by the review team in each situation.

For imprecision, the evidence was downgraded one level if the sample size was 50–100 individuals. If the sample size was less than 50 individuals the evidence was downgraded two levels.

As this review included studies in which the >80% of the population had a chronic diagnosis, the risk of bias associate with indirectness did not exist and therefore the evidence was not downgraded for indirectness.

## Results

The initial search yielded 9,906 articles, of which 7,377 remained after duplicates were removed. Five articles met the inclusion criteria and were deemed eligible for inclusion in the review. Fig 1 outlines the search results and screening process using a PRISMA flow diagram [29].

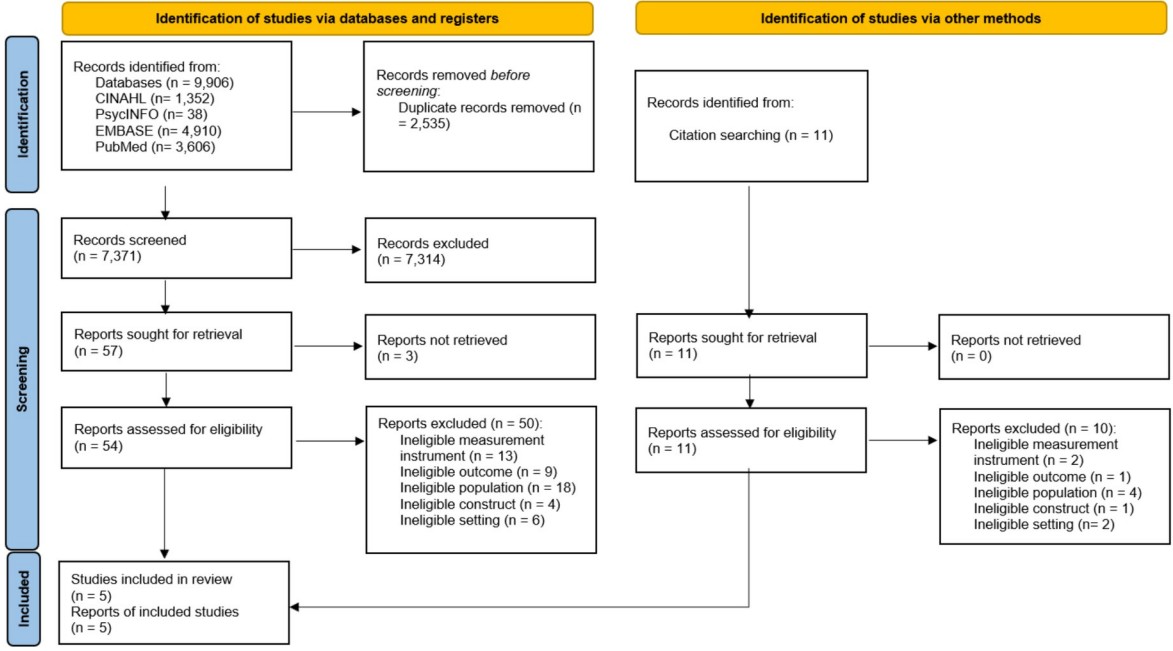

**Fig 1. Process of identification, screening and exclusion of studies according to the PRISMA statement [29].**

## Study characteristics

A summary of the descriptive characteristics of each included study and the included perfor-
mance-based measures is presented in Table 1. Four of the included studies were conducted in
Australia [36–39], with the remaining study conducted in the United States [40]. A total of 77
individuals were included in the review with sample sizes ranging from 10–26 participants. Two
studies included patients with Parkinson's Disease [38, 39] and the remaining three studies
included patient cohorts with stroke, cystic fibrosis and heart failure respectively [36, 37, 40].

Measurement properties of 15 performance-based physical function measures were investi-
gated in the included studies. Inter-rater and intra-rater reliability were reported for 12 of the
measures, while the criterion validity of all 15 measures was reported. No studies reported data
on measurement error or responsiveness.

Of the 15 performance-based measures, seven assessed balance (Timed Up and Go Test,
functional and lateral reach tests, Berg Balance Scale, step test, steps in 360 degree turn, and
timed stance test) [36, 39, 40], two assessed exercise capacity (3 minute step test and 6 minute
walk test) [36, 37] and two assessed functional strength (grip and pinch strength) [36, 38]. The
remaining four measures assessed diverse aspects of functional capacity including the Func-
tional Independence Measure (FIM) [38], of which only the motor components were assessed
(bathing, dressing, toileting, walking, stairs, eating, grooming, bladder management, toilet
transfers, bowel management bed/chair transfers, tub/shower transfers), the Unified Parkin-
son's Disease Rating Scale (UPDRS) [38], of which relevant items were assessed (posture, gait,
sensory complaints, falling, freezing when walking, tremor, tremor at rest, salivation, facial
expression, bradykinesia, speech, action or postural hand tremor, handwriting), the Nine Hole
Peg Test (38) and the European Stroke Scale [40].

## Methodological quality of included studies

The COSMIN risk of bias scores for the measurement properties of the measures in each
included study are displayed in Table 2. Of the studies that reported the reliability of the
included measures, ten measures demonstrated adequate quality while three demonstrated
inadequate quality. Of the studies that reported criterion validity of the measures, eight mea-
sures demonstrated very good quality and ten demonstrated inadequate quality. Many of the
studies reporting the criterion validity of the included measures received an inadequate quality
rating as per COSMIN guidance as the correlation with the reference standard was not calcu-
lated [41] (e.g. mean differences between measures administered via telehealth compared to
face to face administration were reported as opposed to correlations).

## Overall rating and quality of evidence

A summary of the overall rating and quality of evidence per measurement property of the
included measures is presented in Table 3. These scores were developed from the information
displayed in Table 2 which included the rating and the COSMIN risk of bias score. Twelve
measures received 'sufficient' overall ratings for reliability, with a 'very low' quality of evidence
score. Eight measures received 'sufficient' overall ratings and seven received 'indeterminate'
ratings for criterion validity and were all scored as 'low' or 'very low' quality of evidence. For
example, the Six Minute Walk Test (6MWT) demonstrated sufficient reliability (ICC>0.70)
with a 'very low' quality of evidence score due to the inadequate COSMIN risk of bias rating of
the included study and the small sample size (n<50). The 6MWT also demonstrated 'suffi-
cient' validity (correlation with face-to-face>0.70) with a 'low' quality of evidence score due to
the 'very good' COSMIN risk of bias rating of the included study and the low sample size
(n<50). The 3-minute step test demonstrated 'indeterminate' validity as no correlation with

**Table 1. Characteristics of included studies.**

| Study | Country | Performance Measure | Physical Function Domains | Telehealth Environment | Equipment Required | Participant Population | Mean age ± SD (range) | Rater Population | Measurement Properties Assessed |
|---|---|---|---|---|---|---|---|---|---|
| **Cox et al. 2013** [37] | Melbourne, Australia | 3-minute step test | Exercise capacity | Administered using synchronous videoconferencing platform by clinician in separate room to the participant within the same building | 15cm high step, metronome, pulse oximeter | N = 10 adults with cystic fibrosis recruited prospectively on admission to hospital, N = 5 males, N = 5 females, mean FEV1 = 55.4% of predicted (range = 38–90% of predicted) | 32 years ± 7 years | Not reported | Criterion validity |
| **Hoffmann et al. 2008** [38] | Queensland, Australia | FIM (motor components), UPDRS (selected items), Nine Hole Peg Test, Grip strength Pinch strength | Functional strength, functional capacity | Administered using synchronous videoconferencing platform by clinician in a separate room to the participant | Hand-held dynamometer, Pinch gauge | N = 12 community-dwelling participants with Parkinson's Disease, adequate cognitive status to participate in assessment tasks, N = 6 males, N = 6 females, N = 6 tested in telehealth, N = 6 tested face-to-face | 66.1 years ± 8.5 years | N = 3 assessors | Inter-rater reliability, intra-rater reliability, criterion validity |
| **Hwang et al. 2017** [36] | Brisbane, Australia | TUGT, 6MWT, grip strength | Functional balance and mobility, exercise capacity, functional strength | Administered using synchronous videoconferencing by clinician in a separate room within the same hospital building | TUGT: stopwatch, 45cm high chair with arm rests, 3m walk track, regular footwear ± mobility aid6MWT: 30m track, stopwatch, automatic sphygmomanometer, finger pulse oximeter, lap counter Grip strength: Hand-held dynamometer | N = 17 patients with chronic heart failure, 88% males, 12% females | 69 years ± 12 years | N = 4 hospital physiotherapists with an average of 11.5 years of work experience in physiotherapy | Inter-rater reliability, intra-rater reliability, criterion validity |
| **Palsbo et al. 2007** [40] | United States of America | European Stroke Scale, Functional Reach Test | Functional capacity, functional balance | Administered using synchronous videoconferencing platform by clinician in a separate room to the participant | European Stroke Scale: examination table Functional Reach Test: large yard stick | N = 26 patients with a history of stroke including both inpatients and outpatients, N = 18 males, N = 18 females, time since stroke range 2 months-15 years, mean = 2.7 years | Median age = 64 years | N = 4 physiotherapists from rehabilitation hospitals, all had at least 2 years of experience using telehealth to support onsite physiotherapists for a variety of patient assessments | Criterion validity |
| **Russell et al. 2013** [39] | Queensland, Australia | TUGT, step test, steps in 360 degree turn, timed stance test, Berg Balance Scale, lateral reach test, functional reach test | Functional balance and mobility | Administered using synchronous videoconferencing platform by clinician in a separate room to the participant | TUGT: stopwatch Lateral and Functional Reach Tests: calibrated assessment tool | N = 12 people with Parkinson's Disease, adequate cognitive status to participate in assessment tasks, N = 6 males, N = 6 females, mean age at time of diagnosis = 53.5 years, SD = 9.0, range = 38–69 years, average number of years since diagnosed with Parkinson's = 6.8 years, SD = 4.4, range 2–15 years | 66.1 years ± 8.5 years (45–76 years) | N = 1 final year physiotherapy and N = 2 occupational therapy students | Inter-rater reliability, intra-rater reliability, criterion validity |

FIM = Functional Independence Measure, UPDRS = Unified Parkinson's Disease Rating Scale, TUGT = Timed Up and Go Test, 6MWT = 6 Minute Walk Test, FEV1 = Forced Expiratory Volume in one second, N = sample size, SD = standard deviation; cm = centimetres, m = metres

**Table 2. Measurement properties of performance-based measures.**

| Performance-based Measure | Reliability | | | | | Criterion Validity | | |
|---|---|---|---|---|---|---|---|---|
| | Result | Design | Time Interval | COSMIN Risk of Bias Score | Overall Rating | Result | COSMIN Risk of Bias Score | Overall Rating |
| **Exercise Capacity** | | | | | | | | |
| 6MWT [36] | $ICC_{2,1}$ >0.99 $ICC_{1,1}$ >0.99 | Inter-rater Intra-rater | Same day | Inadequate Inadequate | + + | $ICC_{1,1}$ (95%CI) 0.90 (0.74–0.96) MD (95%CI) 4 (-25 to 17 metres) | Very good | + |
| 3 min Step Test [37] | | | | | | MD lowest SpO2 0.2% (LoA -3.4 to 3.6%), MD rate of perceived exertion 0.5 points (LoA -1.1 to 2.1 points) MD heart rate -0.6 beats/min (LoA -11.3 to 10.1 beats/min) | Inadequate | ? |
| **Strength Tests** | | | | | | | | |
| Grip Strength [38] | | | | | | Authors report "no differences" observed | Inadequate | ? |
| Grip Strength [36] | $ICC_{2,1}$ >0.99 $ICC_{1,1}$ >0.99 | Inter-rater Intra-rater | Same day | Inadequate Inadequate | + + | Right hand: $ICC_{1,1}$ (95%CI) 0.94 (0.84–0.98) Left hand: $ICC_{1,1}$ (95%CI) 0.96 (0.89–0.98) | Very good | + + |
| Pinch Strength [38] | | | | | | Authors report "no differences" observed | Inadequate | ? |
| **Balance Tests** | | | | | | | | |
| Berg Balance Scale [39] | $ICC_{2,1}$≥0.96 $ICC_{2,1}$≥0.98 | Inter-rater Intra-rater | 2 months | Adequate Adequate | + + | Kappa 0.94, %EA 16.7, %A ±1 75.0 | Very good | + |
| TUGT [36] | $ICC_{2,1}$ 0.95 (0.86–0.98) $ICC_{1,1}$ 0.96 (0.90–0.99) | Inter-rater Intra-rater | Same day | Inadequate Inadequate | + + | $ICC_{1,1}$ (95%CI) 0.85 (0.64–0.94) MD (95%CI) 0.24 (-0.56 to 1.03) seconds | Very good | + |
| TUGT [39] | $ICC_{2,1}$≥0.96 $ICC_{2,1}$≥0.98 | Inter-rater Intra-rater | 2 months | Adequate Adequate | + + | LoA 1.25 to 1.24 Clinically acceptable limit 5.00 MD -0.01, SD 0.63 MAD 0.47 | Inadequate | ? |
| Step Test [39] | $ICC_{2,1}$≥0.96 $ICC_{2,1}$≥0.98 | Inter-rater Intra-rater | 2 months | Adequate Adequate | + + | Right foot: Kappa 0.97, %EA 75.0, %A ±1 83.3 Left foot: Kappa 0.95, %EA 66.7, %A ±1 83.3 | Very good Very good | + + |
| Functional Reach Test [39] | $ICC_{2,1}$≥0.96 $ICC_{2,1}$≥0.98 | Inter-rater Intra-rater | 2 months | Adequate Adequate | + + | LoA -2.71 to 0.69 Clinically acceptable limit 4.74 MD -1.01, SD 0.87 MAD 1.01 | Inadequate | ? |
| Functional Reach Test [40] | | | | | | No significant difference between results (Z = -0.239, p>0.05) 92% of participants scored within 95% agreement limits | Inadequate | ? |
| Steps in 360 degrees turn [39] | $ICC_{2,1}$≥0.96 $ICC_{2,1}$≥0.98 | Inter-rater Intra-rater | 2 months | Adequate Adequate | + + | Right foot: Kappa 0.98, %EA 75.0, %A ±1 100.0 Left foot: Kappa 0.97, %EA 66.7, %A ±1 91.7 | Very good Very good | + + |
| Lateral Reach Test [39] | $ICC_{2,1}$≥0.96 $ICC_{2,1}$≥0.98 | Inter-rater Intra-rater | 2 months | Adequate Adequate | + + | MD -0.79, SD 0.66, LoA -2.09 to 0.51, clinically acceptable limit 4.74, MAD 0.82 | Inadequate | ? |

(*Continued*)

**Table 2.** (Continued)

| Performance-based Measure | Reliability | | | | | Criterion Validity | | |
|---|---|---|---|---|---|---|---|---|
| | Result | Design | Time Interval | COSMIN Risk of Bias Score | Overall Rating | Result | COSMIN Risk of Bias Score | Overall Rating |
| Timed Stance Test [39] | ICC$_{2,1}$≥0.96 ICC$_{2,1}$≥0.98 | Inter-rater Intra-rater | 2 months | Adequate Adequate | + + | LoA -4.17 to 5.06, clinically acceptable limit 8.00, MD 0.44, SD 2.35, MAD 1.58 | Inadequate | ? |
| **Functional Capacity Tests** | | | | | | | | |
| FIM (motor components) [38] | ICC$_{2,1}$ 0.95 ICC$_{2,1}$ 0.94 | Inter-rater Intra-rater | 1 week 2 months | 1 week 2 months | + + | Kappa 0.93, %EA 91.6%, %A ±1 98.7% | Very good | + |
| UDPRS (selected items) [38] | ICC$_{2,1}$ 0.80 ICC$_{2,1}$ 0.84 | Inter-rater Intra-rater | 1 week 2 months | Adequate Adequate | + + | Kappa 0.81, %EA 73.4%, %A ±1 95.2% | Very good | + |
| European Stroke Scale [40] | | | | | | No significant difference between results (Z = -0.239, p>0.05) | Inadequate | ? |
| Nine Hole Peg Test [38] | ICC$_{2,1}$ 0.99 ICC$_{2,1}$ 0.99 | Inter-rater Intra-rater | 1 week 2 months | Adequate Adequate | + + | Right hand: MD 0.25 seconds (SD 0.90), LoA -2.02 to 1.52, MAD 0.68 seconds Left hand: MD 0.14 seconds, SD 0.61, LoA -1.34 to 1.05, MAD 0.45 seconds | Inadequate Inadequate | ? ? |

FIM = Functional Independence Measure, UPDRS = Unified Parkinson's Disease Rating Scale, TUGT = Timed Up and Go Test, 6MWT = 6 Minute Walk Test, %

EA = Percent exact agreement, %A ±1 = Percent agreement within one point on ordinal scale, SD = Standard deviation, MAD = Mean absolute difference,

ICC = intraclass correlation coefficient, MD = Mean difference, LoA = Limits of agreement, + = sufficient rating,? = indeterminate rating

the reference standard (face-to-face administration) was reported, which was insufficient information reported to provide a 'sufficient' rating according to the COSMIN guidance. The summary scores for the validity of the Timed Up and Go Test (TUGT) and grip strength reflect adjustments that were made to allow for inconsistencies in the results reported in the included studies. For example, the validity of the TUGT was reported in two studies and received an overall 'sufficient' validity rating (correlation>0.70) [36], with 'very low' quality of evidence due to the inadequate COSMIN risk of bias score of the included studies, the small sample size (<50), and the inconsistency between the validity findings of the included studies [36–39].

## Exercise capacity measures

Measures of exercise capacity included in this review were the 6MWT and three minute step test. The 6MWT demonstrated sufficient reliability and criterion validity when administered via telehealth. Evidence for the administration of three minute step test via telehealth is yet to be determined as there was no information available examining its reliability, and the evidence for criterion validity was indeterminate due to non-optimal analysis. Therefore recommendation for the use of this instrument via telehealth cannot be made. However, the mean differences between the telehealth assessment and the face-to-face assessment observed by Cox et al. [37] were very small and suggest that there was no significant difference between the telehealth assessment and face-to-face administration. Therefore these three minute step test results are encouraging.

**Table 3. Summary of findings.**

| Reliability | Summary Result | Overall Rating | Quality of Evidence |
|---|---|---|---|
| 6MWT | ICC>0.99; sample size: 17 | Sufficient | Very Low (one inadequate study, sample <50–100) |
| Step Test | ICC≥0.96; sample size: 12 | Sufficient | Very Low (one adequate study, sample <50–100) |
| Grip Strength | ICC>0.99; sample size: 17 | Sufficient | Very Low (one inadequate study, sample <50–100) |
| Berg Balance Scale | ICC≥0.96; sample size: 12 | Sufficient | Very Low (one adequate study, sample <50–100) |
| TUGT | ICC>0.95; total sample size: 29 | Sufficient | Very Low (multiple studies of at least inadequate quality, sample <50–100, consistent results) |
| Functional Reach Test | ICC≥0.96; sample size: 12 | Sufficient | Very Low (one adequate study, sample <50–100) |
| Steps in 360 degree turn | ICC≥0.96; sample size: 12 | Sufficient | Very Low (one adequate study, sample <50–100) |
| Lateral Reach Test | ICC≥0.96; sample size: 12 | Sufficient | Very Low (one adequate study, sample <50–100) |
| Timed Stance Test | ICC≥0.96; sample size: 12 | Sufficient | Very Low (one adequate study, sample <50–100) |
| FIM (motor components) | ICC range 0.94–0.95; sample size: 12 | Sufficient | Very Low (one adequate study, sample <50–100) |
| UDPRS (selected items) | ICC range 0.80–0.84; sample size: 12 | Sufficient | Very Low (one adequate study, sample <50–100) |
| Nine Hole Peg Test | ICC 0.99; sample size: 12 | Sufficient | Very Low (one adequate study, sample <50–100) |
| **Criterion Validity** | | | |
| 6MWT | ICC 0.90, mean difference of 4; sample size: 17 | Sufficient | Low (one very good study, sample <50–100) |
| Step Test | Kappa range 0.95–0.97, %EA ≥66.7, %A ±1 83.3; sample size: 12 | Sufficient | Low (one very good study, sample <50–100) |
| 3 min Step Test | MD Sp02 0.2%, MD rate of perceived exertion 0.5 points, MD heart rate -0.6 beats/min; sample size: 10 | Indeterminate | Very Low (one inadequate study, sample <50–100) |
| Grip Strength | Right hand: $ICC_{1,1}$ (95%CI) 0.94 (0.84–0.98) Left hand: $ICC_{1,1}$ (95%CI) 0.96 (0.89–0.98); authors report "no differences" observed; total sample size: 29 | Sufficient | Very Low (multiple studies of at least inadequate quality, sample <50–100, inconsistent results) |
| Pinch Strength | Authors report "no differences" observed; sample size: 12 | Indeterminate | Very Low (one inadequate study, sample <50–100) |
| Berg Balance Scale | Kappa 0.94, %EA 16.7, %A±1 75.0; sample size: 12 | Sufficient | Low (one very good study, sample <50–100) |
| TUGT | ICC 0.85, MD 0.24 seconds, LoA 1.25 to 1.24, CAL 5.00, MD -0.01, SD 0.63; total sample size: 29 | Sufficient | Very Low (multiple studies of at least inadequate quality, sample <50–100, inconsistent results) |
| Functional Reach Test | LoA -2.71 to 0.69, CAL 4.74, MD -1.01, SD 0.87, MAD 1.01; No significant difference between results (Z = -0.239, p>0.05), 92% of participants scored within 95% agreement limits; total sample size: 29 | Indeterminate | Very Low (multiple studies of at least inadequate quality, sample <50–100, consistent results) |
| Steps in 360 degrees turn | Kappa range: 0.97–0.98, %EA≥66.7, %A ±1 ≥ 91.7; sample size: 12 | Sufficient | Low (one very good study, sample <50–100) |

(*Continued*)

**Table 3.** (Continued)

| Reliability | Summary Result | Overall Rating | Quality of Evidence |
|---|---|---|---|
| Lateral Reach Test | MD -0.79, SD 0.66, LoA 2.09 to 0.51, CAL 4.74, MAD 0.82; sample size: 12 | Indeterminate | Very Low (one inadequate study, sample <50–100) |
| Timed Stance Test | LoA -4.17 to 5.06, CAL 8.00, MD 0.44, SD 2.35, MAD 1.58; sample size: 12 | Indeterminate | Very Low (one inadequate study, sample <50–100) |
| FIM (motor components) | Kappa 0.93, %EA 91.6, %A±1 98.7; sample size: 12 | Sufficient | Low (one very good study, sample <50–100) |
| UDPRS (selected items) | Kappa 0.81, %EA 73.4, %A±1 95.2; sample size: 12 | Sufficient | Low (one very good study, sample <50–100) |
| European Stroke Scale | No significant difference between results (Z = -0.239, p>0.05); sample size: 26 | Indeterminate | Very Low (one inadequate study, sample <50–100) |
| Nine Hole Peg Test | MD range 0.14–0.25 seconds, SD range 0.61–0.90, MAD range 0.45–0.68seconds; sample size: 12 | Indeterminate | Very Low (one inadequate study, sample <50–100) |

MD = Mean difference, MAD = Mean Absolute Difference, ICC = Intraclass correlation coefficient, CAL = Clinically acceptable limits, LoA = Limits of agreement %EA = Percent exact agreement, %A±1 = Percent agreement within one point on ordinal scale, SD = Standard deviation, FIM = Functional Independence Measure, UPDRS = Unified Parkinson's Disease Rating Scale, TUGT = Timed Up and Go Test, 6MWT = 6 Minute Walk Test

## Functional strength measures

The grip strength test demonstrated sufficient reliability and criterion validity. Evidence for the pinch strength measure administered via telehealth is yet to be determined due to the lack of information available.

## Functional balance measures

Seven measures of functional balance were included in the review. Measures with the most robust results demonstrating sufficient reliability and criterion validity were the Berg Balance Scale, TUGT, Step test and the Steps in 360 degree turn test. The other measures (Functional Reach Test, Lateral Reach Test and Timed Stance Test) all demonstrated sufficient reliability, however, the criterion validity of these measures administered via telehealth when compared to face-to-face administration could not be determined due to non-optimal analysis. The mean difference between the telehealth and face-to-face administration of the functional reach test of -1.01 as observed by Russell et al. [39] lies within the limits of agreement of -2.71 to 0.69. This is also within the clinically acceptable limit of 4.74cm [42], supporting telehealth administration of the functional reach test [43]. Similarly, the mean difference observed between the telehealth and face-to-face administration of the lateral reach test [39] of -0.79 is within the reported limits of agreement (-2.09 to 0.51) and clinically acceptable limit (4.74cm) [42], which supports this measure being administered via telehealth. Finally, the mean difference of 0.44 [39] between the timed stance tests when administered via telehealth compared to face-to-face administration is also within the limits of agreement (-4.17 to 5.06), and is less than the clinically acceptable limit (8.00 seconds). Therefore it can be reasonably assumed that the telehealth administration of the timed stance test is valid when compared to face-to-face administration.

## Functional capacity measures

Other measures included in the review which measured various aspects of general functional capacity included the European Stroke Scale, Unified Parkinson's Disease Rating Scale,

Functional Independence Measure and Nine Hole Peg Test. The most robust results were reported for the Functional Independence Measure and the Unified Parkinson's Disease Rating Scale which both demonstrated sufficient reliability and criterion validity. However, as the Unified Parkinson's Disease Rating Scale is a population-specific measure, the Functional Independence Measure may be more appropriate for various chronic populations. While the Nine Hole Peg Test demonstrated sufficient reliability, the criterion validity was indeterminate due to non-optimal analysis. However, the mean differences of 0.25 seconds (right hand) and 0.14 seconds (left hand) observed between telehealth administration and face-to-face administration of the measure were both within the limits of agreement of -2.02 to 1.52 (right hand) and -1.34 to 1.05 (left hand), which is encouraging. Evidence for the reliability and criterion validity of the European Stroke Scale administered via telehealth is yet to be determined due to the lack of information available and non-optimal analysis.

## Discussion

This systematic review identified five studies which examined the psychometric properties of fifteen performance-based measures of physical function administered via telehealth among people with various chronic conditions. Overall, there is low-very low evidence demonstrating sufficient reliability and criterion validity for a range of measures across each domain of exercise capacity, strength, balance and functional capacity when administered via telehealth and compared to face-to-face administration. The overall quality of evidence was low-very low, reflecting the small number of studies, the small sample sizes of the included studies and non-optimal analyses (i.e. failure to correlate scores with the reference standard face-to-face administration method) as per the COSMIN risk of bias tool.

The findings of sufficient reliability and criterion validity when administered via telehealth mirror that reported when many measures are administered among chronic populations in a face-to-face environment, including the 6MWT [44, 45] and the grip strength test [46, 47], as well as the Berg Balance Scale, TUGT and Step Test [48–52].

As per COSMIN guidance, in order to demonstrate 'sufficient' validity the measure administered via telehealth must demonstrate >0.70 correlation with the measurement when administered in a face-to-face setting. While the included measures which appeared to be valid when compared to face-to-face administration, the correlations were not calculated and therefore there was not sufficient information to classify the criterion validity as 'sufficient' as per the COSMIN standards. Also, the quality of the included studies were downgraded for this same reason. Therefore these findings should be interpreted with caution.

Although the included studies did not report on all measurement properties for each measure, sufficient evidence was reported for the reliability and criterion validity of some measures across several domains. No evidence regarding the measurement error or responsiveness of the included measures was reported in the included studies.

### Strengths and limitations

Strengths of this review include the prospective protocol registration, following PRISMA guidance, as well as using two reviewers for screening, shortlisting and data extraction. A particular strength is using the COSMIN approach, which had not been used in previous psychometric evaluations for performance measures via telehealth.

There were also some limitations to be acknowledged. As previously stated, two independent reviewers screened a sample of 30% of abstracts and relevant full texts to determine eligibility. As good agreement was achieved on the 30% sample, the remaining screening process was not performed in duplicate. Although the quality of the summarised evidence was rated

using the modified GRADE approach by two independent reviewers, neither of these reviewers were formally trained in the use of this method. Due to the heterogeneous nature of the included measures and the populations of the included studies, a meta-analysis could not be performed and the results could not be quantitatively summarised. As the majority of the results could not be combined, best evidence synthesis was mostly obtained from a single study. Further evidence may have been identified from studies of post-operative populations, such as individuals post total knee arthroplasty [53]. However, these were excluded as pain and disability levels immediately post-operatively, and how much these change relatively quickly, are quite different from other chronic conditions where physical function may be more stable over time. There was limited information reported regarding the characteristics of the samples in the included studies in relation to aspects such as socioeconomic status, cognitive status and technological literacy. Also, the included studies were all carried out in countries with 'very high' Human Development Index scores [54]. These factors could potentially impact the external validity of the findings. Although Cox et al. [37] reported the usability of the three minute step test administered via telehealth and Hwang et al. [36] reported some information regarding the number and nature of technical issues encountered during telehealth administration of the 6MWT, TUGT and grip and pinch strength, there was limited information reported in the included studies regarding the interpretability and feasibility of the included measures when administered via telehealth. As noted in the eligibility criteria, this review was concerned with examining the validity of measures administered via telehealth when compared to face-to-face administration of the same measure. Therefore, other types of validity, such as content and construct validity, were not reported in this review. Due to the eligibility criteria and aims of this review, the outcomes of interest were reliability, measurement error and criterion validity. For this reason, we followed COSMIN recommendations for evaluating reliability, measurement error and criterion validity but a priori did not choose to evaluate other types of validity, internal structure, interpretability and feasibility.

## Clinical implications

Encouragingly, several performance-based measures of physical function across different domains (e.g. exercise capacity, strength and balance) may have satisfactory reliability and criterion validity when used in a telehealth environment. Furthermore, the psychometric properties of these measures appears similar to that reported for the same measures when used in a face-face context. This should reassure clinicians that using performance-based measures of physical function via telehealth is possible. However, this evidence is of low-very low quality and there is a significant lack of information regarding the measurement error and responsiveness of these measures. Furthermore, information regarding the interpretability and feasibility of the included measures was very limited [36, 37].

## Future research

This systematic review highlights the need for further larger, high quality research, in line with COSMIN guidance, exploring the psychometric properties of performance-based measures of physical function administered via telehealth among people with various chronic conditions. In particular more studies examining the measurement error, responsiveness, interpretability and feasibility of these instruments are required. Although some of the measures included in this review demonstrated sufficient reliability and validity, none of the measures were evaluated with respect to all measurement properties and therefore strong recommendations cannot yet be made. Additionally, the lack of studies exploring the administration of performance-based measures of physical function via telehealth among chronic musculoskeletal populations

is acknowledged. Therefore this review highlights the need for future studies to be conducted in this population.

## Conclusion

A wide range of performance-based measures measuring various domains of physical function administered via telehealth among chronic populations have been identified in this review. All of these measures appear to be reliable when used in a telehealth environment. Validity of these measures is less certain, and there is no information regarding the measurement error or responsiveness of these measures. Further high quality research is required to examine the psychometric properties of a core set of measures administered via telehealth among people with chronic health conditions, particularly regarding measurement error, responsiveness, feasibility and interpretability.

## Supporting information

**S1 Fig. Preferred reporting items for systematic reviews and meta-analyses (PRISMA) guidelines.**
(PDF)

**S2 Fig. Search strategy.**
(PDF)

**S1 Table. Criteria for good measurement properties.**
(PDF)

**S2 Table. Instructions on the use of the modified GRADE approach.**
(PDF)

## Acknowledgments

We acknowledge the support of Liz Dore, Faculty of Education and Health Sciences librarian at the Glucksman Library in the University of Limerick, for her assistance with the development of the search strategy.

## Author Contributions

**Conceptualization:** Caoimhe Barry Walsh, Roisin Cahalan, Rana S. Hinman, Kieran O' Sullivan.

**Formal analysis:** Caoimhe Barry Walsh.

**Supervision:** Roisin Cahalan, Kieran O' Sullivan.

**Validation:** Roisin Cahalan, Kieran O' Sullivan.

**Visualization:** Caoimhe Barry Walsh.

**Writing – original draft:** Caoimhe Barry Walsh.

**Writing – review & editing:** Roisin Cahalan, Rana S. Hinman, Kieran O' Sullivan.

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
