## [Decision Letter · Decision Letter 0]

15 Jun 2022

PONE-D-22-14057Psychometric properties of performance-based measures of physical function administered via telehealth among people with chronic conditions: A systematic reviewPLOS ONE

Dear Dr. Barry Walsh,

Thank you for submitting your manuscript to PLOS ONE. After careful consideration, we feel that it has merit but does not fully meet PLOS ONE’s publication criteria as it currently stands. Therefore, we invite you to submit a revised version of the manuscript that addresses the points raised during the review process.

Additional Editor Comments:

Please find the comments of the reviewers below. All of the reviewers indicated essential major and minor revisions. King Regards

We look forward to receiving your revised manuscript.

Kind regards,

Fatih Özden, PhD

Academic Editor

PLOS ONE

Journal Requirements:

Reviewers' comments:

Reviewer's Responses to Questions

**Comments to the Author**

1. Is the manuscript technically sound, and do the data support the conclusions?

Reviewer #1: Partly

Reviewer #2: Yes

Reviewer #3: Partly

Reviewer #4: Yes

2. Has the statistical analysis been performed appropriately and rigorously? 

Reviewer #1: Yes

Reviewer #2: I Don't Know

Reviewer #3: Yes

Reviewer #4: Yes

3. Have the authors made all data underlying the findings in their manuscript fully available?

Reviewer #1: Yes

Reviewer #2: Yes

Reviewer #3: Yes

Reviewer #4: Yes

4. Is the manuscript presented in an intelligible fashion and written in standard English?

Reviewer #1: Yes

Reviewer #2: Yes

Reviewer #3: Yes

Reviewer #4: Yes

5. Review Comments to the Author

Reviewer #1: Thank you for the opportunity to review this manuscript. The study represents an important piece of work evaluating measurement properties for measurement tools for physical function using the COSMIN checklist. The review appears to be robust and well conducted, and provides a clear summary of the reliability measurements undertaken in the literature.

I found the description of the methods and findings around validity less clear and would suggest some significant modifications before this study is suitable for publication.

1) Use of the COSMIN Checklist: the references to the COSMIN checklist link to recommendations for patient reported outcome measures. The use of COSMIN methodology seems reasonable for these performance based (not self reported) tools, but needs to be justified in the methods and discussed in the study limitations

2) The COSMIN user manual that is referenced recommends a 10 step process for systematic review of measurement properties. For a comprehensive review such as this one I would expect statements describing the methodology and findings for:

*Content validity including tool development

*Internal structure

*Remaining measurement properties (reliability, measurement error, criterion validity, hypothesis testing + responsiveness)

*Interpretability and feasibility

While some of these domains may not be directly applicable for the tools you have identified, and others may have no available data, I would suggest this should be acknowledged in your methods and then considered in your discussion as deviations from the COSMIN method.

3) Study construct / measurement instrument selection: you describe the construct of interest as "performance-based physical function" and your measurement instruments of interest as "performance measures". Do you mean that the type of physical function you are interested in relates to performance, or that the measurement tools require an evaluation of performance (i.e. not self reported). I would suggest that you use performance-based for one of these uses but not both.

4) Evaluation of measurement properties. It would be helpful to have a description of the validity/reliability domains and the thresholds applied to decide if measurement properties of satisfactory/indeterminant and inadequate to guide a reader who is unfamiliar with COSMIN through your methods

5) 'Gold standard': The authors have used a gold standard of fact to face measurement for criterion validity. I would argue that this is more likely to be a reliability measurement, or even a hypothesis test of construct validity as it is possible that an invalid test for physical function may yield similar values if administered face to face or via telemedicine. While this is an important measurement property I do not feel that it is sufficient robust to use as gold standard.

6) Study characteristics: I would be interested to know where the patients were located for these validation studies. Were these patients at home (akin to telemedicine during the pandemic) or patients brought to a clinic or laboratory to participate in the study. This would have significant feasibility implications for patient-at-home telemedicine rehabilitation as discussed in the background.

7) Methodological quality of studies: I found this section unclear as study quality (risk of bias) seemed to blend with the adequacy of measurement properties reported. In addition, I am confused as to how studies could be selected for reporting criterion validity but then downgraded for not reporting correlation. How did these studies compare their telemedicine to face to face measurements?

8) Summary of findings: You report unidimensionality in this table but this has not been described in the background or methods. How dis you assess unidimensionality and is this relevant for single-item scores?

9) Discussion lines 323-377 are a well written description of the results and I feel would be better situated in your results section.

10) Clinical implications: Given the very low quality of evidence available I do not think you can conclude that any measures have adequate reliability and validity in this study. I think the implication should be toned down to several performance measures may have satisfactory reliability and validity.

11) Strengths and limitations: you discuss limitations in the studies included but not the potential limitations and mitigations from your study design and execution. These include:

Use of the COSMIN checklist for non-PROM tools

Not undertaking study screening and full text extraction in duplicate

Not seeking indirect evidence of measurement properties from other populations

Reviewer #2: Dear authors, I would like to congratulate you for your article that will contribute to the literature.

A few things I can say about the study:

1- Correct the " COSMIN (Consensus-Based Standards for the Selection of Health Measurement Instrument)" phrase in lines 28 and 29 to " Consensus-Based Standards for the Selection of Health Measurement Instrument (COSMIN)". Correct other spellings like this throughout the manuscript.

2- Indicate which authors are the reviewers mentioned on lines 163-169.

3- Write the reference representations of the studies mentioned in Table 1 and Table 2.

4- Specify the definitions of abbreviations such as n, SD in Table 1.

Reviewer #3: The manuscript that aimed to the investigate psychometric properties of performance-based

measures of physical function administered via telehealth among people with chronic

health conditions using the COSMIN (Consensus-Based Standards for the Selection of

Health Measurement Instrument) approach. Overall, the study is well written. Authors should address the following concerns:

1- Search strategy should be updated.

2- Abstract: authors should report specific chronic conditions, telehealth environment and tools tested in the results and make conclusion on the quality of the current evidence supporting any investigate tool, if so. Besides, it is important to raise the potential issue related to the external validity (only high Human Development Index/HDI settings?).

3- Methods, selection of studies: Only 30% of abstracts and potential full texts were independently assessed by two independent reviewers in the selection of studies. Add it as a limitation in the discussion section.

4- Methods, page 10. Please add reference for the following statement: “As per COSMIN guidance, in order to demonstrate ‘sufficient’ validity the measure must

198 demonstrate >0.70 correlation with the ‘gold standard’.”.

5- Methods, synthesis: please clarify if it is appropriate to include studies not comparing with face to face.

6- Methods: criteria do downgrade que quality of the evidence in each domain of the GRADE approach should be specified. Was it conducted by two independent and trained reviewers? How did they resolve potential disagreements, if so?

7- Results: further description on characteristics of the samples is needed because it may be a potential external validity to be discussed. Participants: from the community? rural?, Socioeconomic status?, Cognitively tested?, and able to use mobile, computer, both?

8- Results, Table 1: Hwang et al 2017 conducted synchronous videoconferencing as all the other included studies? Please clarify in the Table.

9- Discussion: authors should revise discussion section to accommodate comments.

10- References: replace reference 29 to “LB Mokkink, M Boers, CPM van der Vleuten, LM Bouter, J Alonso, DL Patrick, HCW de Vet, CB Terwee. COSMIN Risk of Bias tool to assess the quality of studies on reliability or measurement error of outcome measurement instruments: a Delphi study. BMC Medical Research Methodology. 2020;20(293).”. Is is the specific reference for performance-based outcome measures (PerFOMs).

11- References: reference for risk of bias should be “Lidwine B. Mokkink, Maarten Boers, CPM van der Vleuten, LM Bouter, Jordi Alonso, Donald L Patrick, HCW de Vet, CB Terwee. COSMIN Risk of Bias tool to assess the quality of studies on reliability or measurement error of outcome measurement instruments: a Delphi study. BMC Medical Research Methodology. 2020;20(293).” Besides, please revise if it was used correctly by two independent reviewers with disagreements resolved by consensus or a third reviewer.

Reviewer #4: PONE-D-22-14057

Title: Psychometric properties of performance-based measures of physical function

administered via telehealth among people with chronic conditions: A systematic review

Thank you for the opportunity to review the above-mentioned manuscript. This article aims to examine the psychometric properties of performance-based measures of physical function administered via telehealth among people with chronic health conditions. I agree with the authors as they mention in the introduction, the necessity of such reviews for measures of chronic conditions. I would like to congratulate the authors for such impressively high-quality work. I absolutely enjoyed reading this work. I have only a few minor comments.

Abstract-

Results: please re-order according to frequency

What about other types of validity such as content, face, and construct validity? Can you add a bit about these too?

Also, you discuss reliability, which type of reliability was that?

Introduction-

This section was extremely well-written.

Line 59- “The ageing nature of our population..” I agree this is true in many Western countries but not necessarily everywhere, therefore, I suggest changing our.

Methods-

Authors did a meticulous work in this section as well.

Line 142- for the constructs, please add a few examples

Line 154- "However, since the comparator was always face-to-face administration of the same measure, when extracting data from the selected studies the only relevant properties were reliability, measurement error and criterion validity."

This is great explanation but can go into your discussion or limitation. I think here under measurement properties you must say that you planned to include any measurement property reported in the papers.

Line 160- I think you can connect using just 'or' instead of and/or. This is discouraged in academic writing.

Results-

Line 252- minor grammatical error, use were instead of was.

Line 254- I suggest adding something like ‘…Or other types of validity’

Line 255- Same comment as in the abstract, please re-order based on the frequency of the tests, or alphabetically.

Discussion-

Line 326- Consider re-writing: This evidence reflects the evidence supporting…Line 379- remove 'of' in following of PRISMA

Line 400- I would be cautious to say that '....have satisfactory reliability and validity when used in a telehealth environment.' especially with validity since you only found data for criterion validity and other types were not addressed.

6. PLOS authors have the option to publish the peer review history of their article (what does this mean?). If published, this will include your full peer review and any attached files.

Reviewer #1: No

Reviewer #2: No

Reviewer #3: No

Reviewer #4: No

---

## [Author Response · Author response to Decision Letter 0]

28 Jul 2022

We would like to thank you for giving us the opportunity to submit a revised version of the manuscript and express our thanks to the Reviewers for their constructive feedback and helpful suggestions. We believe that these revisions in response to the comments made by the Reviewers have resulted in an improved manuscript. 

We have uploaded our specific responses to each of the comments made by the reviewers in the file titled 'Response to Reviewers'.

---

## [Decision Letter · Decision Letter 1]

26 Aug 2022

Psychometric properties of performance-based measures of physical function administered via telehealth among people with chronic conditions: A systematic review

PONE-D-22-14057R1

Dear Dr. Barry Walsh,

We’re pleased to inform you that your manuscript has been judged scientifically suitable for publication and will be formally accepted for publication once it meets all outstanding technical requirements.

Kind regards,

Fatih Özden, PhD

Academic Editor

PLOS ONE

Additional Editor Comments (optional):

Reviewers' comments:

Reviewer's Responses to Questions

**Comments to the Author**

1. If the authors have adequately addressed your comments raised in a previous round of review and you feel that this manuscript is now acceptable for publication, you may indicate that here to bypass the “Comments to the Author” section, enter your conflict of interest statement in the “Confidential to Editor” section, and submit your "Accept" recommendation.

Reviewer #1: All comments have been addressed

Reviewer #2: All comments have been addressed

Reviewer #3: All comments have been addressed

Reviewer #4: All comments have been addressed

2. Is the manuscript technically sound, and do the data support the conclusions?

Reviewer #1: Yes

Reviewer #2: Yes

Reviewer #3: Yes

Reviewer #4: Yes

3. Has the statistical analysis been performed appropriately and rigorously? 

Reviewer #1: Yes

Reviewer #2: Yes

Reviewer #3: Yes

Reviewer #4: Yes

4. Have the authors made all data underlying the findings in their manuscript fully available?

Reviewer #1: Yes

Reviewer #2: Yes

Reviewer #3: Yes

Reviewer #4: Yes

5. Is the manuscript presented in an intelligible fashion and written in standard English?

Reviewer #1: Yes

Reviewer #2: Yes

Reviewer #3: Yes

Reviewer #4: Yes

6. Review Comments to the Author

Reviewer #1: Thank you for asking me to review this manuscript again. I have read the new submission and found all comments to be addressed

Reviewer #2: (No Response)

Reviewer #3: Authors have clarified all comments and manuscript has good quality to be published. I believe the readers will be interested.

Reviewer #4: I have no further comments. The requested revisions in my previous round were fully satisfied. The manuscript is technically sound and is in good shape for publication.

7. PLOS authors have the option to publish the peer review history of their article (what does this mean?). If published, this will include your full peer review and any attached files.

Reviewer #1: No

Reviewer #2: No

Reviewer #3: No

Reviewer #4: No

---

## [Editor Report · Acceptance letter]

30 Aug 2022

PONE-D-22-14057R1 

Psychometric properties of performance-based measures of physical function administered via telehealth among people with chronic conditions: A systematic review 

Dear Dr. Barry Walsh:

I'm pleased to inform you that your manuscript has been deemed suitable for publication in PLOS ONE. Congratulations! Your manuscript is now with our production department. 

Kind regards, 

on behalf of

Dr. Fatih Özden 

Academic Editor

PLOS ONE